# Salinity Mitigates the Negative Effect of Elevated Temperatures on Photosynthesis in the C_3_-C_4_ Intermediate Species *Sedobassia sedoides*

**DOI:** 10.3390/plants13060800

**Published:** 2024-03-12

**Authors:** Elena Shuyskaya, Zulfira Rakhmankulova, Maria Prokofieva, Nina Lunkova, Pavel Voronin

**Affiliations:** K.A. Timiryazev Institute of Plant Physiology of Russian Academy of Science, 127276 Moscow, Russia; zulfirar@mail.ru (Z.R.); maria.vdovitchenko@gmail.com (M.P.); nina.lunkova@gmail.com (N.L.); pavel@ippras.ru (P.V.)

**Keywords:** combined stress, cyclic electron transport around photosystems I, C_2_ CCM, malate valve, ROS

## Abstract

The adaptation of plants to combined stresses requires unique responses capable of overcoming both the negative effects of each individual stress and their combination. Here, we studied the C_3_-C_4_ (C_2_) halophyte *Sedobassia sedoides* in response to elevated temperature (35 °C) and salinity (300 mM NaCl) as well as their combined effect. The responses we studied included changes in water–salt balance, light and dark photosynthetic reactions, the expression of photosynthetic genes, the activity of malate dehydrogenase complex enzymes, and the antioxidant system. Salt treatment led to altered water–salt balance, improved water use efficiency, and an increase in the abundance of key enzymes involved in intermediate C_3_-C_4_ photosynthesis (i.e., Rubisco and glycine decarboxylase). We also observed a possible increase in the activity of the C_2_ carbon-concentrating mechanism (CCM), which allowed plants to maintain high photosynthesis intensity and biomass accumulation. Elevated temperatures caused an imbalance in the dark and light reactions of photosynthesis, leading to stromal overreduction and the excessive generation of reactive oxygen species (ROS). In response, *S. sedoides* significantly activated a metabolic pathway for removing excess NADPH, the malate valve, which is catalyzed by NADP-MDH, without observable activation of the antioxidant system. The combined action of these two factors caused the activation of antioxidant defenses (i.e., increased activity of SOD and POX and upregulation of *FDI*), which led to a decrease in oxidative stress and helped restore the photosynthetic energy balance. Overall, improved PSII functioning and increased activity of PSI cyclic electron transport (CET) and C_2_ CCM led to an increase in the photosynthesis intensity of *S. sedoides* under the combined effect of salinity and elevated temperature relative to high temperature alone.

## 1. Introduction

Global climate change, which is characterized by warming and aridification, may intensify the impact of various forms of environmental stress on plants, which themselves may become more acute, widespread, and may have differential effects on plant species when present in combination. Moreover, this may lead to the exacerbation of desertification and soil salinization. The intensification of stressors often leads to growth inhibition, damage to photosynthetic processes, and ultimately to lower crop quality and yield [1]. Studies have indicated that plant responses to combined stresses may be unique and, in some cases, antagonistic to individual stresses [2]. Plant adaptation to combined stresses therefore requires a specific combined response involving physiological, biochemical, and molecular reactions that are specially organized to overcome the negative effects of each individual stressor as well as any emergent properties of their combination [3].

Photosynthesis is highly sensitive to high-temperature stress and can be impaired before other cellular functions are disrupted. The main targets of high-temperature stress include photosystem II (PSII) and ribulose-1,5-bisphosphate carboxylase/oxygenase (Rubisco). However, most plants, regardless of their photosynthetic type, have a significant ability to adapt their photosynthetic processes in response to the temperature of the ambient environment, a process known as temperature acclimation. Increasing growth temperatures lead to higher optimal photosynthetic temperatures and makes the photosynthetic apparatus more resilient to heat stress [4]. This increased resilience is achieved through the optimization of temperature-sensitive systems, including the oxygen-evolving complex present in PSII, ATP generation, and carbon fixation by Rubisco. Moreover, this also includes an increase in cyclic electron transport (CET) activity around photosystem I (PSI) to sustain ATP synthesis [5]. In addition, high temperatures also lead to the hypergeneration of reactive oxygen species (ROS), including singlet oxygen in PS II and superoxide radical in PS I, among others [4].

For optimal photosynthesis, maintaining cellular energy balance, which involves balancing the production and consumption ratio of ATP to NADPH, is required [6,7]. Various environmental stressors can disrupt this balance, leading to ROS production in both photosystems [8]. Plants have various photoprotective mechanisms that maintain energy balance, one of which is nonphotochemical quenching (NPQ). NPQ mechanisms increase the proportion of absorbed light energy in PSII that is dissipated as heat, thereby protecting PSII from photodamage by singlet oxygen [8,9]. In addition, several alternative electron flow pathways are known to support ATP synthesis but not NADPH synthesis. These include two cyclic electron transport pathways around PSI, i.e., the PGR5/PGRL1 protein-dependent pathway and the NADH dehydrogenase-like complex-dependent pathway. Although PGR5/PGRL1 is considered the main CET pathway in C_3_ plants, both pathways contribute to CET and can partially compensate for each other’s activity [7,10,11]. PSI CET increases the ATP/NADPH production ratio in the electron transport chain (ETC), thereby avoiding NADPH production while maintaining ΔpH accumulation for ATP synthesis [5,12,13]. It has been shown that an excess of ATP can inhibit both CET pathways, but the PGR5/PGRL1 pathway is two to three times more sensitive to this kind of suppression [14]. Moreover, an increase in the stromal NADPH/NADP+ ratio can activate the malate valve. The key component of the chloroplast malate valve is the NADPH-dependent enzyme MDH, which oxidizes stromal NADPH while reducing OAA to malate. Malate is then exported to the cytosol in exchange for OAA [7,15,16]. Thus, the PSI CET and malate valve are considered to be the most important additional pathways used to maintain the chloroplast energy balance [7,17].

One of the initial plant responses to salinity stress is a reduction in the rate of photosynthesis associated with a limitation in CO_2_ availability, suppression of photosynthetic enzymes, and decreases in the efficiency of photosystems and the structural integrity of the thylakoid membrane [18,19,20]. The negative effects of nonstomatal limitations have been attributed to ion imbalances and increased ROS production in the leaves [18,21]. Moreover, an increase in Na^+^ content in the leaves can lead to dose-dependent inhibition of PSII activity [22]. However, Na^+^ toxicity alone cannot explain all of the negative effects of salinity stress on photosystem efficiency [19]. Salt-tolerant plants known as halophytes have specific mechanisms to overcome the osmotic and ionic effects of salinity and the associated oxidative stress. The transport and binding of sodium ions, the synthesis and transport of compatible solutes, and the rapid activation of antioxidant defenses all play crucial roles in salinity tolerance [23,24,25]. One important mechanism of salt tolerance involves early root signaling, which includes K^+^ and H^+^ transport, Ca^2+^ signaling, non-selective cation channels (NSCC), phospholipid modifications, ROS induction, and protein kinase activity [25,26]. However, in general, the early cellular responses to salt related to Na^+^ import and sensing are currently poorly understood [25]. A positive feedback relationship exists between ROS and Ca^2+^, where Ca^2+^ induces ROS production and ROS induces Ca^2+^ import and promotes K^+^ efflux via the stomata. In addition, Ca^2+^ may induce Na^+^ efflux via several calcineurin B-like proteins-interacting protein kinases (CBL-CIPK) pathways, including the salt overly sensitive (SOS) pathway. Calcium is sensed by SOS3/CBL4, which binds to SOS2/CIPK24, and the resulting complex phosphorylates the H^+^/cationic antiporter SOS1/NHX7, which can transport Na^+^ out of the cell. Finally, CBL10 has been found to form a complex with SOS2, which plays an important role in Na^+^ sequestration in vacuoles [25,27]. 

The species *Sedobassia sedoides* (Chenopodiaceae) has been classified as a C_3_-C_4_ intermediate species with C_2_ type of photosynthesis based on its anatomical features, gas exchange analysis, and the immunolocalization of glycine decarboxylase (GDC) [28]. C_2_ photosynthesis is a carbon concentration mechanism (CCM) that reassimilates CO_2_ released via photorespiration [29,30,31]. C_2_ photosynthetic activity depends on the photorespiratory decarboxylation of glycine via the GDC complex in bundle sheath cells. C_2_ metabolism is hypothesized to provide plants with several advantages over C_3_ species, including an expanded ecological niche in warmer and drier areas, improved net carbon assimilation (via CCM) and stress tolerance. Despite these benefits, relatively little is known regarding the C_2_ pathway relative to other photosynthetic types, especially under stress conditions [31]. The aim of this study was therefore to investigate the impact of elevated temperature (35 °C) and salinity (300 mM NaCl), both individually and in combination, on the functioning of the photosynthetic apparatus of the C_3_-C_4_ (C_2_) halophyte *S. sedoides*.

## 2. Results

### 2.1. Biomass, Water, and Na^+^ and K^+^ Content

The effect of salinity on *S. sedoides* seedlings did not lead to a decrease in the fresh and dry biomass of shoots, while their water content decreased by 20% (Figure 1A–C). Growing the seedlings at elevated temperatures led to a decrease in dry biomass by 26% and an increase in water content by 25% (Figure 1B,C). As a result of the combined effects of elevated temperature and salinity, a reduction in fresh biomass by 35% was observed, although the accumulation of dry biomass did not significantly differ from the control (Figure 1B). 

The K^+^ content in the shoots remained unchanged under the applied factors, while the accumulation of Na^+^ by *S. sedoides* plants increased under all treatments, most significantly (by 2.5 times) under the combined effect of elevated temperature and salinity, leading to a decrease in the K^+^/Na^+^ ratio under all treatments (Figure 1C–E).

### 2.2. CO_2_/H_2_O Gas Exchange

Salinity did not cause changes in the intensity of apparent photosynthesis (*A*) in *S. sedoides* plants. However, *A* decreased by six-fold under elevated temperature and by two-fold under the combined effect of elevated temperature and salinity (Figure 2A). Transpiration intensity (*E*) decreased by ~1.5-fold under all treatments (Figure 2B). Water use efficiency (WUE) increased under salinity, but significant decreased under elevated temperature (Figure 2C). Dark respiration (*Rd*) decreased by two-fold under elevated temperature (Figure 2D).

### 2.3. Activity of PSI CET and PSII Efficiency

Salinity led to an increase in the PSI CET activity by 40%, as indicated by the time required to reach the maximum P700 oxidation level under far-red light (Figure 3A). Conversely, when plants were grown at elevated temperature, the PSI CET activity, maximum quantum yield (*F*_v_/*F*_m_), and effective quantum yield in the light-adapted sate (*F’*_v_/*F’*_m_) of PSII significantly decreased (Figure 3B,C). Under the combined effect of elevated temperature and salinity, the reduction in these parameters was less compared with elevated temperature-only treatment (Figure 3). At the same time, there was no increase in nonphotochemical quenching (NPQ) (Figure 3D).

### 2.4. Photosynthetic Enzymes Content

Salinity resulted an increase in Rubisco content by 35% in *S. sedoides*, while elevated temperature and the combined effect of the two factors caused a nearly two-fold decrease (Figure 4A,B). The content of GDC decreased by two-fold under elevated temperature but increased under salinity and the combined effect of the two factors (Figure 4A,C). No significant changes in the PEPC content were identified under all treatments (Figure 4A,D).

### 2.5. Expression of Photosynthetic Genes

Transcript accumulation analysis was performed for several genes, including *rbcL* (encoding the large subunit of Rubisco), *GLDP* (encoding the P subunit of GDC), *Ppc2* (encoding PEPC), *PPDK* (encoding pyruvate, orthophosphate dikinase), *psaA* and *psaB* (encoding apoproteins of PSI), *psbA* (encoding the D1 protein of PSII), *CAB* (encoding chlorophyll *a*/*b*-binding protein LHCB/CAB PSII), *FDI* (encoding ferredoxin I), *PGR5* (encoding the PGR5 protein, a key component of the main CET pathway of PSI), and *NdhH* (encoding the ndhH subunit of the NADH dehydrogenase in the second CET pathway of PSI) (Figure 5). Salinity resulted in a decrease in *PGR5* expression. Elevated temperature treatment enhanced the expression of *rbcL* and *psbA*, while the combined effect of elevated temperature and salinity upregulated the expression of *rbcL*, *PPDK*, and *FDI* and downregulated *PGR5* expression (Figure 5).

### 2.6. Lipid Peroxidation and Antioxidant Enzyme Activity

MDA content increased by 42% under salinity and more than doubled under elevated temperature. Under the combined effect of elevated temperature and salinity, the level of MDA was similar to that under salt treatment alone (Figure 6A). SOD activity decreased under elevated temperature but increased under the combined effect of elevated temperature and salinity (Figure 6B). The simultaneous action of both factors also led to an increase in POX activity, while CAT activity remained relatively unchanged (Figure 6C,D).

### 2.7. Enzyme Activity of the Malate Dehydrogenase Complex

Under salinity conditions, regardless of the growth temperature, there was approximately a two-fold decrease in the activity of NAD-MDH (Figure 7A). Elevated temperature resulted in a four-fold increase in the activity of NADP-MDH and a 50% increase in NADP-ME (Figure 7B). The combined action of both factors led to a decrease in the activity of NADP-ME (Figure 7C). The activity of NAD-ME enzyme remained unchanged under all treatments (Figure 7D).

### 2.8. Multifactor Analysis Using Principal Component Analysis (PCA)

PCA divided *S. sedoides* plants into three groups (Figure 8). Plants grown under elevated temperature were clearly separated from the others by the first principal component (PC1), which accounted for 41.14% of the total variation. The main elements of PC1 were the NADP-MDH activity, *F’*_v_/*F’*_m_, GDC content, Rd and A intensity, WUE, and MDA levels. Plants subjected to combined stress were separated from the others by the second principal component (PC2). The main elements of PC2 were Na^+^ content, K^+^/Na^+^ ratio, and POX and NAD-MDH activity. Both principal components were sufficient to explain 67.27% of the changes in the total variation.

## 3. Discussion

### 3.1. Effect of Salinity 

Under natural conditions, the most productive *S. sedoides* plants were found in habitats with a soil salinity of ~90 mM NaCl/kg soil [32]. In previous model experiments, long-term treatment with 200 mM NaCl was found to decrease plant biomass in some *S. sedoides* populations [33,34]. In our experiment, a four-day treatment with 300 mM NaCl caused decreases in transpiration intensity and water content due to the osmotic component of salinity [35]. However, this did not lead to decreases in the intensity of photosynthesis and the dry biomass accumulation of *S. sedoides* (Figure 1 and Figure 2). This result was probably due to the sodium ion concentration, which is not critical for *S. sedoides* (2.12 ± 0.31 mmol Na^+^/g DM) (Figure 1). In natural habitats, and over a range of different salinities, *S. sedoides* maintains tissue Na^+^ concentrations within the range of 2.1–2.7 mmol/g DM [32,36]. Thus, halophytes like *S. sedoides* may require this Na^+^ concentration for optimal growth [18,24].

We also found that salinity caused slight oxidative stress, as evidenced by a 42% increase in MDA content (Figure 6). However, the stable maximum quantum yield of PSII is indicative of the activity of photoprotective mechanisms in *S sedoides*. It has previously been shown that CET around PSI can play an important role in protecting photosystems from damage caused by stromal overreduction [5,37]. Here, the enhanced activity of PSI CET in *S. sedoides* (Figure 3) may help prevent photoinhibition of photosystems in response to ROS overproduction, as was evident for the CAM species *Jatropa curcas* under saline conditions [38]. At the same time, the CCM of intermediate C_3_-C_4_ species, as well as in C_4_ and CAM plants, requires high amounts of ATP for CO_2_ fixation. PSI CET generates additional ATP, contributing to the establishment of a pH gradient across the thylakoid membrane without the NADPH formation [39,40]. In this regard, the enhanced activity of the PSI CET in *S. sedoides* could indirectly indicate an increase in C2 CCM activity (Figure 9). Additional indirect evidence includes downregulation of the *PGR5* gene, which encodes the PGR5 protein of PGR5/PGRL1 pathway. In C_3_ plants, the PGR5/PGRL1 pathway is the primary PSI CET pathway, while NDH-dependent PSI CET plays a significant role in C_4_ photosynthesis [5,7,10,11]. A decrease in *PGR5* expression may diminish the role of PGR5/PGRL1-dependent CET in favor of the NDH-dependent CET of PSI. Here, an increase in C_2_ CCM changes the NADH/NAD balance since NADH is formed during glycine oxidation. It is known that in C_3_ plants, NADH accumulation can inhibit some key tricarboxylic acid flux (TCA) enzymes, including NAD-MDH [41]. While in C_2_ species, it has been suggested that TCA may function as a non-cycling pathway under conditions of light and photorespiration [31]. In our experiment, we observed a decrease in NAD-MDH activity in *S. sedoides* in response to salt treatment (Figure 7).

Salinity stress also led to an increase in the content of Rubisco and GDC in *S. sedoides*. However, this did not occur as a result of increased expression of the corresponding *rbcL* and *GLDP1* genes (Figure 5). The discrepancy between gene expression and protein content may be due to changes in the balance between protein synthesis and degradation [42,43], although imbalances in gene expression and protein content may also be a consequence of posttranscriptional regulation [44,45,46].

Overall, we found that salinity caused additional Na^+^ accumulation and decreased transpiration, which led to changes in the water–salt balance and caused increased water use efficiency. Furthermore, higher C_2_ CCM activity was observed due to increased PSI CET activity with a possible shift in the PGR5/NdhH ratio toward the NDH CET pathway and increased Rubisco and GDC content (Figure 9). These changes made it possible to maintain high levels of photosynthesis activity and biomass accumulation, i.e., comparable to the level of control plants. Moreover, the absence of a negative effect of salinity was also evidenced by multivariate analysis (PCA), which did not identify a clear separation between control plants and plants treated with NaCl (Figure 8).

### 3.2. Effect of Elevated Temperatures

Elevated temperature treatment led to lower Rubisco protein content and the upregulation of rbcL expression in *S. sedoides* (Figure 4 and Figure 5), indicating that there was a significant imbalance between protein synthesis and degradation. The decrease in the content of the main photosynthetic enzymes (i.e., Rubisco and GDC, Figure 4) indicates the suppression of metabolic reactions associated with photosynthesis. This in turn may cause an imbalance between dark and light reactions and excessive generation of ROS [7]. For example, with decreasing Calvin cycle activity, the consumption of NADP+ decreases, which in turn causes the formation of superoxide anion radicals (O_2_•) associated with incomplete oxidation of water (in PSII) or the oxidation of ferredoxin (in PSI) [47,48]. In *S. sedoides*, an imbalance in the stromal NADPH/NADP^+^ ratio was indicated by a significant increase in malate valve activity (i.e., a four-fold increase in NADP-MDH activity, Figure 7). In the malate valve, stromal NADPH is consumed by the reduction of oxaloacetate (OAA) to malate, which is then exported to the cytosol in exchange for OAA [7,15] (Figure 9). Moreover, NADP-MDH activity is subject to strict redox regulation, and the malate valve is active only when the stromal NADPH/NADP^+^ ratio is high [15]. Another additional electron flow that supports ATP synthesis but not NADPH synthesis is CET around PSI [7]. The PSI CET can therefore put upward pressure on the ATP/NADPH production ratio by maintaining the accumulation of ΔpH for ATP synthesis without triggering NADPH synthesis [7,40]. However, elevated temperature decreased CET PSI activity in *S. sedoides*, in contrast to the salinity, which led to increased CET PSI activity (Figure 3 and Figure 9).

PSII was salinity tolerant in *S. sedoides*, but elevated temperatures led to a significant decrease in PSII efficiency (Figure 3). In general, photosystem II is considered to be more sensitive to heat and oxidative stress than salinity especially in halophytes [4,49]. This is because excess ROS can inactivate the elongation factor G for D1 proteins; therefore, its synthesis can be inhibited at the same time PSII activity is lost [50]. Increased expression of the *psbA* gene (Figure 5), which encodes the D1 protein, may be a response to decreasing D1 content, which in turn leads to a decrease in the effective and maximum quantum yield of PSII in *S. sedoides* (Figure 3). Another reason for the decreased PSII efficiency of *S. sedoides* may be the fact that nonphotochemical quenching (NPQ) levels remained unchanged (Figure 3). This unit dissipates the light energy absorbed by PSII as heat, protects PSII from photodamage by singlet oxygen, and slows down the ETC electron flow rate [8,9]. In addition, a decrease in dark respiration was observed in *S. sedoides* (Figure 2), which is not a typical plant response to high temperatures [51,52]. However, a decrease in dark respiration due to insufficient protection from ROS has been observed in some salt-tolerant species [53].

Severe oxidative stress in *S. sedoides* (i.e., a two-fold increase in MDA) and the stable activity levels of antioxidant enzymes (Figure 6) indicate the ineffectiveness of antioxidant protection at high temperatures. It has been hypothesized that antioxidant defenses occur faster in halophytes than in glycophytes [23]. In particular, an excess of sodium ions can disrupt the stability and permeability of cell and organelle membranes, causing complications in energy metabolism and ROS production in various cellular compartments [23]. In *S. sedoides*, high temperatures increased water content and Na^+^ accumulation (i.e., approximately two-fold compared to the control, Figure 1) in the shoots. This accumulation occurred under conditions in which salt stress was absent for the roots (because the 50% Hoagland solution contains less than 0.13 mM Na^+^/L). However, early salt signaling in root cells is currently considered to be an important mechanism of salt tolerance [25,26]. Early signaling responses involve, among others, ROS induction and Ca^2+^ signaling, which activate protein kinase pathways, including salt overly sensitive (SOS) (one of the BL-CIPKs). Here, SOS plays a key role both in the transport of sodium ions out of the cell (SOS1/NHX7) and/or into the vacuole (SOS2), thereby maintaining ion homeostasis [25,27]. Perhaps, the extremely low sodium levels in the root environment did not cause early signaling, which in turn did not cause the rapid activation of the antioxidant enzyme systems that are thought to be characteristic of halophytes. In addition, the “nonactivation” of the SOS regulatory pathway may lead to disturbances in the intracellular transport of sodium ions and their excessive accumulation in the cytosol and in apoplasts. This can cause partial closure of the stomata and therefore decreased transpiration in *S. sedoides* (Figure 2 and Figure 9). It is assumed that when the ability of tissues to accumulate salts in cell vacuoles is exceeded or impaired, an increase in Na^+^ concentrations in the apoplasts around guard cells causes partial closure of the stomata and reduces transpiration [54,55].

Thus, in response to elevated temperatures, we observed a strong decrease in photosynthesis intensity and biomass accumulation of *S. sedoides*. Elevated temperature caused an imbalance between the light and dark photosynthetic reactions, which led to stromal overreduction (i.e., excess NADPH) and excessive generation of ROS (oxidative stress) (Figure 9). To restore the energy balance, the malate valve (catalyzed by NADP-MDH) was significantly activated, and the activity of NADP-ME, the nonphotosynthetic form of which is involved in various protective reactions, also increased [56]. At the same time, the activity of other electron sinks (i.e., NPQ and the PSI CET, among others) and the activity of antioxidant enzymes did not change or decrease. This may be due to ionic imbalances in the shoots/roots (i.e., significant Na^+^ accumulation in the shoots but no salt stress for roots). Thus, these results reflect the “nonactivation” of early salt signaling in the roots, i.e., rapid compartmentalization of ions within vacuoles and the rapid activation of the antioxidant enzyme system.

### 3.3. Combination of the Effects of Elevated Temperature and Salinity

We found that the combined effect of elevated temperature and salinity did not exert more negative effect on the accumulation of dry biomass of *S. sedoides* and that photosynthesis level was higher than that under individual temperature effect (Figure 1 and Figure 2). The contribution of high temperatures to the combined effect on *S. sedoides* plants was determined by quantifying a decrease in Rubisco protein content and the upregulation of *rbcL*; this effect was observed in both the combined and elevated temperature-only treatment (Figure 4). In addition, the observed decrease in the effective and maximum quantum yield of PSII in response to the combined stress treatment was significantly smaller than that observed in the elevated temperature-only treatment. That is, the negative effect of elevated temperature on the functioning of PSII was mitigated by a combination of factors (Figure 3, Figure 7 and Figure 9). Increased salinity-induced thermotolerance of PSII has been identified in the halophytes *Atriplex centralasiatica* [57] and *Suaeda salsa* [49]. Interestingly, in both species, the observed increase in thermotolerance was independent of the level of salinity (i.e., between 100 and 400 mM NaCl) and may have been the result of osmoprotective mechanisms [49,56]. In addition, a similar improvement in the effective and maximum quantum yield of PSII in response to the combined effect of salinity and elevated temperature stresses has also been reported in tomato [58]. In that paper, the authors attributed this finding to improved protective systems, including increased antioxidant protection. In another study, combining salinity and heat treatment stimulated the antioxidant enzymatic defense system in *Jatropha curcas* [59]. In *S. sedoides*, an increase in antioxidant protection was also observed under combined stress. Specifically, we observed an increase in the activity of SOD and POX, which led to a decrease in lipid peroxidation (MDA) relative to the elevated temperature alone treatment (Figure 6 and Figure 7). An increase in antioxidant protection was also evidenced by the upregulation of *FDI* (Figure 6). Overexpression of Fd may increase plant tolerance to abiotic and biotic stress by reducing the levels of ROS produced by the ascorbate-mediated water–water cycle [60,61].

In contrast to the species *S. sedoides*, studies of the C_3_ species *Chenopodium quinoa* showed that the combined effects of elevated temperature and salinity had a more negative effect compared with the action of each factor alone, leading to high oxidative stress and decreases in biomass, PSII efficiency, and photosynthetic gene expression [62]. At the same time, the study of the closely related C_4_-NADP species *Bassia prostrata* showed that elevated temperature changed the role of sodium and potassium ions, as well as proline, in the mechanisms of its salt tolerance [63].

Overall, salt treatment of *S. sedoides* plants grown at elevated temperatures triggers early root signaling and salt tolerance mechanisms that are characteristic of halophytes. Higher Na^+^ content relative to the elevated temperature alone treatment (Figure 1) did not result in an additional negative effect on photosynthesis. In contrast, we observed an improvement in energy balance—i.e., the restoration of an optimal NADPH/NADP^+^ ratio—as evidenced by a decrease in the activity of NADP-MDH (i.e., the malate valve) to the levels found in control plants. PSI CET activity also recovered to control plant levels, possibly due to a shift in the PGR5/NdhH ratio toward the NDH pathway, as observed in the salinity-only treatment group. Relative to the elevated temperature alone treatment, we observed that the combined effect of two factors exhibited an increase in GDC content and PSI CET activity, possibly indicating an increase in C_2_ metabolism in *S. sedoides* (Figure 9). Finally, reduced activity of NADP-ME, which is involved in the metabolic response of plants to stress [64], also indicated a decrease in stress load under the combined effect of two factors relative to the effect of the elevated temperature alone treatment.

## 4. Materials and Methods

### 4.1. Plant Growth Conditions

Seedlings of *Sedobassia sedoides* (Pall.) Freitag and G. Kadereit (Chenopodiaceae) were grown for 30 days in plastic pots filled with perlite. The seedlings were irrigated with a 50% Hoagland nutrient solution. The rearing experiment was conducted in two separate climate chambers, each equipped with circadian lighting provided by commercial fluorescent white light tubes. The light regime consisted of 8 h dark/16 h light (200 mmol/(m^2^ s) PAR, and temperature was maintained at 25 °C and 35 °C. After a 30-day period, half of the plants in both chambers were subjected to a 4-day treatment with 300 mM NaCl. 

### 4.2. Determination of Water, Na^+^, and K^+^ Content

To determine the water content (W), the leaves were dried at 80 °C until a constant weight was achieved. The calculation was performed using the formula: W (g H_2_O/g dry weight) = (FW − DW)/DW,(1)
where FW represents the fresh biomass and DW represents the dry biomass. 

The content of Na^+^ and K^+^ in the leaves was determined in aqueous extracts from 100 mg of dry sample using a flame photometer FPA-2-01 (Zagorsk Optical-Mechanical Plant, Sergiev Posad, Russia) and expressed in mmol/g DW.

### 4.3. CO_2_/H_2_O Gas Exchange

Net photosynthesis was determined by quantifying CO_2_/H_2_O gas exchange per unit leaf area as previously described [65]. Shoots were placed into a darkened, specialized leaf chamber aerated at a constant rate with an air at temperature of 25 ± 1 °C and a relative air humidity of 65–70% (steady-state regime). To measure CO_2_/H_2_O gas exchange, a single-channel infrared gas analyzer LI-820 (LI-COR, Inc., Lincoln, NE, USA) was used. The sample was illuminated (1200 μE PAR) using a fiber-optic light guide from a KL 1500LCD light source (Schott, Mainz, Germany) in an open system. The difference in gas humidity at the inlet and outlet of the leaf chamber was used for calculation of leaf transpiration (E). The steady-state dark respiration (Rd) was measured after turning off the light. Water use efficiency (WUE) was calculated as A/E.

### 4.4. Efficiency of PSII Function and Activity of Cyclic Electron Transport (CET) around PSI

Changes in the redox potential of P700 chlorophyll were measured by monitoring the optical density of leaves at a wavelength of 820 nm using a dual-wavelength pulse-modulated system ED-P700DW (Heinz-Walz GmbH, Effeltrich, Germany) in combination with a PAM-101 fluorometer (Heinz-Walz GmbH, Effeltrich, Germany) [66]. The kinetics of P700 oxidation were measured under far-red light (720 nm, 17.2 W/m^2^, LED 102-FR, Heinz-Walz GmbH). The level of maximum P700 oxidation was determined by applying a flash from a xenon gas-discharge lamp (50 ms, 1500 W/m^2^; Heinz-Walz GmbH) in the presence of far-red light. Measurements were taken after 20 min of dark adaptation of the experimental plants. The activity of CET in PSI was assessed using the time-dependent changes in the kinetic curve of P700 oxidation in response to far-red illumination from the activation of far-red light until the moment of maximum P700 oxidation [40]. 

The quantum yield of PSII photochemistry in dark-adapted (20 min) leaves was determined using a pulse-amplitude-modulated fluorometer (PAM-101, Heinz-Walz GmbH, Effeltrich, Germany) [67]. During the measurements, the sample was illuminated with weak modulated red light. The output signal from the PAM-101 was processed using an analog-to-digital converter (PDA-100, Heinz-Walz GmbH, Effeltrich, Germany) and displayed on a computer. The maximum quantum yield in the dark-adapted state (*F*_v_/*F*_m_), the maximum quantum yield in the light-adapted sate (*F’*_v_/*F’*_m_), and the nonphotochemical quenching (NPQ) values were calculated according to the following equations: *F*_v_/*F*_m_ = (*F*_m_ − *F*_0_)/*F*_m_,(2)
*F’*_v_/*F’*_m_ = (*F’*_m_ − *F’*_0_)/*F’*_m_,(3)
*NPQ* = (*F*_m_ − *F’*_m_)/*F’*_m_,(4)
where *F*_0_ and *F*_m_ are minimal and maximal fluorescence of a dark-adapted leaf, respectively; *F’*_0_ and *F’*_m_ are the minimum and maximum chlorophyll fluorescence after light adaptation, respectively. Chlorophyll fluorescence was measured at room temperature (25 °C).

### 4.5. Content of Photosynthetic Enzymes

Analysis of the content of ribulose-1,5-bisphophate carboxylase/oxygenase (Rubisco), glycine decarboxylase (GDC), and phosphoenolpyruvate carboxylase (PEPC) proteins was determined using immunoblot analysis with SDS-PAGE and commercial polyclonal antibodies obtained from Agrisera (Vännäs, Sweden) for proteins of large subunit (L) of Rubisco (RbcL, AS03037), glycine decarboxylase P protein (GLDP, AS204370), and PEPC (AS09458) as described previously [68]. The obtained results were expressed relative to the average level for control plants, which was taken as 100%. The analysis was performed at least 3 times.

### 4.6. RNA Isolation and Quantitative Real Time (RT)-PCR

Total RNA was extracted from leaf samples (0.50 g FW) using phenol–chloroform extraction with precipitation using LiCl as described previously [68]. Reverse transcription was performed according to the standard Evrogen protocol (Evrogen, Moscow, Russia). PCR primers were designed using Pick Primers NCBI (National Center for Biotechnology Information, Bethesda, MD, USA) with Primer Pair Specificity Checking Parameters and SnapGene Viewer (4.2.11) on nucleotide sequences available in the NCBI database (Table 1). The transcript levels were assessed with real-time PCR (RT-qPCR) using a Light Cycler96 amplifier (Roche, Basel, Switzerland) with SybrGreen I dye (Evrogen, Moscow, Russia). RT-PCR data were analyzed using Light Cycler96 Software Version 1.1. Relative quantification was performed to compare the levels of the target gene and the reference gene, and the result was expressed as a ratio. *UBQ10* and *b-Tubulin* were used as reference genes. Transcript levels were calculated relative to control plants.

### 4.7. Assay of Antioxidant Enzyme Activity and Lipid Peroxidation

Frozen plant shoots (0.5 g) were homogenized in 0.1 M Tris-HCl (pH 7.4) containing 1 mM dithiothreitol (DTT), 0.5 mM phenylmethylsulfonyl fluoride, and 0.5% dimethyl sulfoxide (DMSO). The homogenates were centrifuged at 10,000× *g* for 15 min at 4 °C. The supernatant was used to determine the activity of antioxidant enzymes. Protein content was determined using the Bradford method with bovine serum albumin (“Dia-M”, Moscow, Russia) as the standard.

Superoxide dismutase (SOD; EC 1.15.1.1) activity was determined using the following reaction mixture: 0.1 M Tris-HCl (pH 7.8), 50 µM nitroblue tetrazolium, 10 mM L-methionine, 0.025% Triton X-100, and 3 µM riboflavin. The reaction was initiated by adding the enzyme extract and exposed to light at an intensity of 350 µmol/(m^2^ s) [69]. Absorbance was recorded at 560 nm (Multiskan™ Sky, “Thermo Fisher Scientific”, Waltham, MA, USA).

Peroxidase (POD; EC 1.11.1.7) activity was determined by recording the increase in optical density at 470 nm during the oxidation of guaiacol [70]. The reaction mixture contained 0.1 M Tris-HCl (pH 7.4), 7 mM guaiacol, 4 mM H_2_O_2_, and the enzyme extract.

Catalase (CAT; EC 1.11.1.6) activity was determined by mixing the enzyme extract with 0.1 M Tris-HCl (pH 7.4) and 0.1 M H_2_O_2_. Catalase activity was measured using the decrease in optical density at 240 nm for 1 min [71]. 

The intensity of lipid peroxidation was determined according to Heath and Packer [72] method by measuring malondialdehyde (MDA) using the reaction with thiobarbituric acid. The optical density of the samples was measured at 532 nm and 600 nm. The MDA content was calculated using the absorption coefficient 155 mM^−1^ cm^−1^.

### 4.8. Assay of Malate Dehydrogenase Enzyme Activity

The activities of NAD(P)^+^-dependent malate dehydrogenases (NAD-MDH and NADP-MDH) and NAD(P)^+^-dependent malic enzymes (NAD-ME and NADP-ME) were determined spectrophotometrically with changes in the optical density of the reaction mixture at 340 nm, indicating the rate of NAD(P)H consumption. A unit of enzyme activity was equal to the enzyme amount that transforms 1 µmol of substrate for 1 min under standard conditions [73]. Total protein was extracted from 0.3 g of frozen plant shoots using a pre-cooled mortar and pestle in 1.5 mL of ice-cold 50 mM Tris-HCl buffer (pH 8.2) containing 10 mM MgCl_2_, 0.3 mM EDTA, 2% (*v*/*w*) polyvinylpyrrolidone, and 5 mM DTT. To determine the rate of oxaloacetate/pyruvate reduction, the reaction medium contained 50 mM Tris-HCl, pH 8.0 (NAD(P)-MDH) or pH 6.5 (NAD(P)-ME); 3 mM oxaloacetate/pyruvate; and 0.2 mM NADH/(NADPH). Here, 10 mM MgCl_2_ and 5 mM MnCl_2_ were used as coenzymes for NAD(P)-MDH and NAD(P)-ME, respectively.

### 4.9. Statistical Analyses

Factor (ANOVA) analyses were carried out using SigmaPlot 12.0 software. The figures show the means and their standard errors. Statistical significance was attained at *p* < 0.05 (Tukey’s test). Multiple factor principal component analysis (PCA) was conducted using R software (version 3.6.1).

## 5. Conclusions

Here, a study of C_3_-C_4_ halophyte *S. sedoides* subjected to elevated temperature, salinity, and combined elevated temperature and salinity treatments revealed weak temperature tolerance in the absence of salinity. *S. sedoides*, which is a halophyte, requires a certain concentration of sodium ions for optimal growth; therefore, changes in the water–salt balance under salinity led to improved water use efficiency and increased content of major enzymes involved in C_3_-C_4_ photosynthesis. Moreover, no negative effects on photosynthesis intensity were observed. Interestingly, salinity, both individual and combined with the elevated temperature treatment, affected parameters associated with C_2_ CCM. This resulted in increased PSI CET activity and a shift in the PGR5/NdhH ratio toward the NDH pathway of CET PSI, as well as increased GDC content. The individual effects of the elevated temperature treatment resulted in significant oxidative stress and decreased biomass accumulation. In general, a sharp decrease in photosynthesis occurs mainly due to decreases in the activity of the Calvin cycle and C_2_ CCM, photosystem efficiency, and an unbalanced energy balance. This may be due to ionic imbalances in the shoots/roots (i.e., significant Na^+^ accumulation in the shoots in the absence of salt stress for roots) and, accordingly, the “nonactivation” of early root signaling, i.e., the rapid compartmentalization of ions in vacuoles and the rapid activation of the antioxidant enzyme system. Overall, the restoration of the photosynthetic energy balance and the improvement in the light and dark photosynthesis reactions of *S. sedoides* plants grown at high temperatures while simultaneously being treated with NaCl suggest the need for early salt signaling to activate protective systems.

## Figures and Tables

**Figure 1 plants-13-00800-f001:**
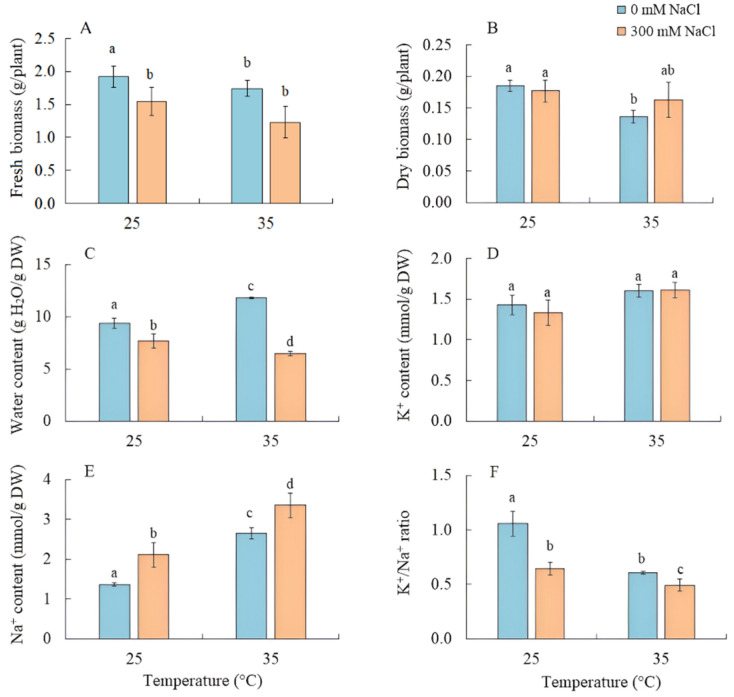
The effect of salinity (300 mM NaCl) and elevated temperature (35 °C) and their interaction with the fresh and dry biomass (**A**,**B**), water (**C**), and K^+^ and Na^+^ (**D**–**F**) contents in *Sedobassia sedoides* shoots. The different letters show statistically different means at *p ≤* 0.05 (Tukey test).

**Figure 2 plants-13-00800-f002:**
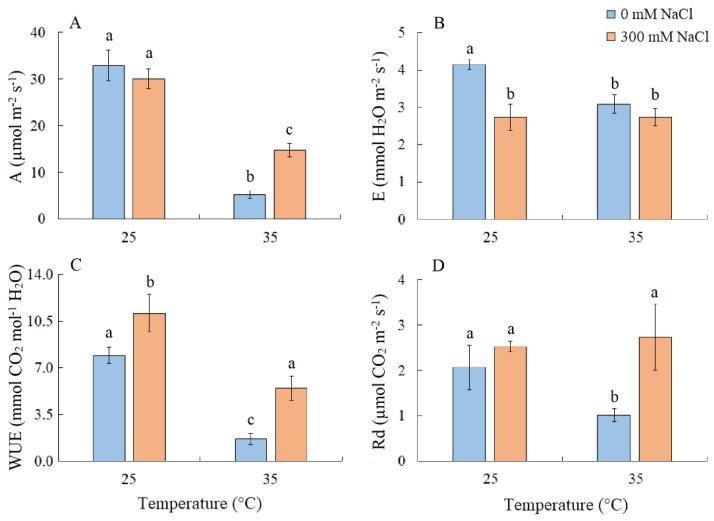
The effect of salinity (300 mM NaCl) and elevated temperature (35 °C) and their interaction with the intensity of apparent photosynthesis (**A**), transpiration (**B**), water use efficiency (**C**), and dark respiration (**D**) in *Sedobassia sedoides* shoots. The different letters show statistically different means at *p* ≤ 0.05 (Tukey test).

**Figure 3 plants-13-00800-f003:**
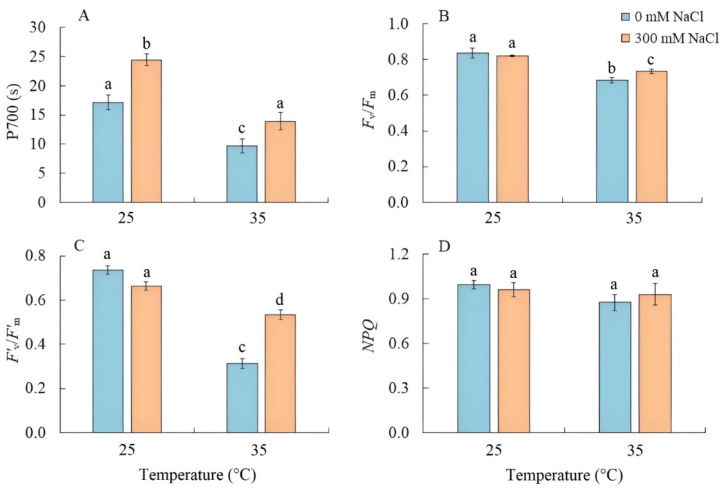
The effect of salinity (300 mM NaCl) and elevated temperature (35 °C) and their interaction with the activity of cyclic electron transport of PSI (**A**), maximum quantum efficiency of PSII (**B**), effective quantum efficiency of PSII photochemistry at a given light intensity (**C**), and nonphotochemical quenching of chlorophyll fluorescence (**D**) in *Sedobassia sedoides* shoots. The different letters show statistically different means at *p ≤* 0.05 (Tukey test).

**Figure 4 plants-13-00800-f004:**
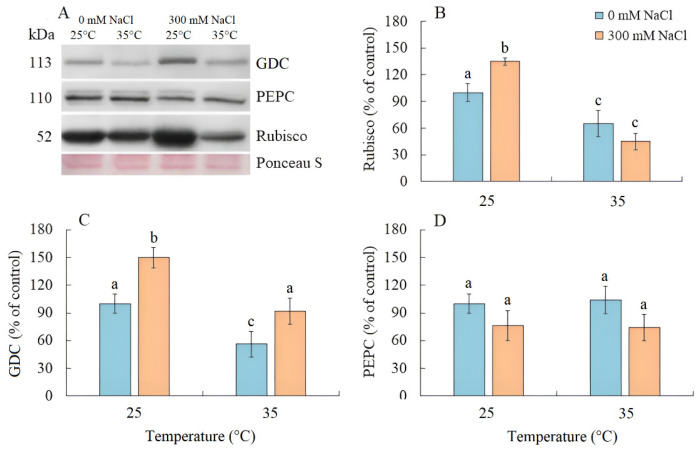
The effect of salinity (300 mM NaCl) and elevated temperature (35 °C), and their interaction with photosynthesis enzymes contents in *Sedobassia sedoides* shoots. (**A**) Western blots for three critical photosynthetic enzymes from total proteins extracted from shoots of *S. sedoides* plants, (**B**) Ribulose-1,5-bisphophate carboxylase/oxygenase (Rubisco, subunit L), (**C**) Glycine decarboxylase (GDC P protein), (**D**) Phosphoenolpyruvate carboxylase (PEPC). Blots were probed with antibodies raised against Rubisco (subunit L), GDC P protein, and PEPC. Relative enzyme contents are shown on the basis of intensity of Western blotting bands estimated using ImageJ 1.37v software (USA) and expressed relative to the average level for control plants taken as 100%. Equal protein loading was checked by staining the blots with Ponceau. Control plants were grown at 25 °C without NaCl treatment. The different letters show statistically different means at *p* ≤ 0.05 (Tukey test).

**Figure 5 plants-13-00800-f005:**
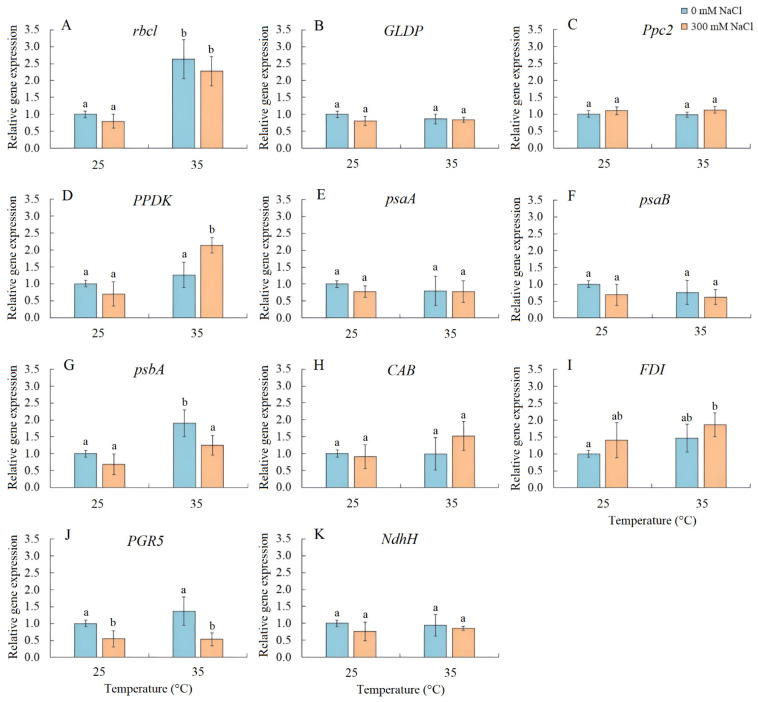
The effect of salinity (300 mM NaCl) and elevated temperature (35 °C) and their interaction with the expression of photosynthetic genes in *Sedobassia sedoides* shoots. (**A**) *rbcL*, encoding the large subunit of Rubisco; (**B**) *GLDP*, encoding the P subunit of GDC; (**C**) *Ppc2,* encoding PEPC; (**D**) *PPDK,* encoding pyruvate, orthophosphate dikinase; (**E**) *psaA* and (**F**) *psaB,* encoding apoproteins of PSI; (**G**) *psbA,* encoding the D1 protein of PSII; (**H**) *CAB,* encoding chlorophyll *a*/*b*-binding protein LHCB/CAB PSII; (**I**) *FDI,* encoding ferredoxin I; (**J**) *PGR5,* encoding the PGR5 protein, a key component of the main CET pathway of PSI; (**K**) *NdhH,* encoding the ndhH subunit of the NADH dehydrogenase in the second CET pathway of PSI. The different letters show statistically different means at *p ≤* 0.05 (Tukey test).

**Figure 6 plants-13-00800-f006:**
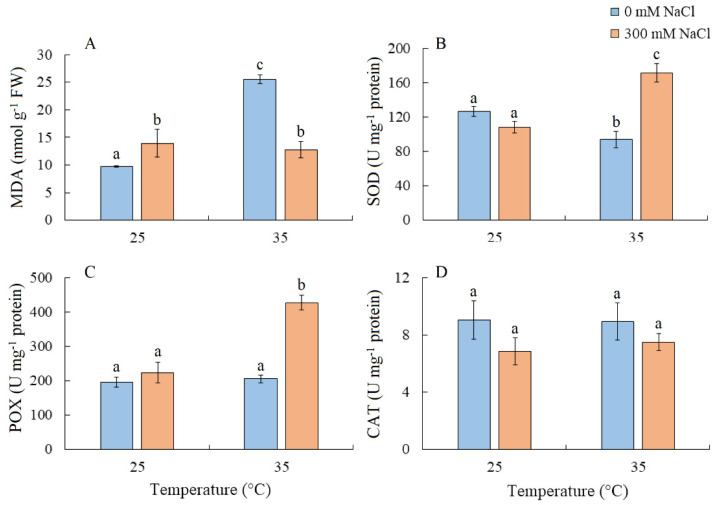
The effect of salinity (300 mM NaCl) and elevated temperature (35 °C) and their interaction with the lipid peroxidation (**A**) and antioxidant enzymes activity (**B**–**D**) in *Sedobassia sedoides* shoots. MDA, malondialdehyde; SOD, superoxide dismutase; POX, peroxidase; CAT, catalase. The different letters show statistically different means at *p ≤* 0.05 (Tukey test).

**Figure 7 plants-13-00800-f007:**
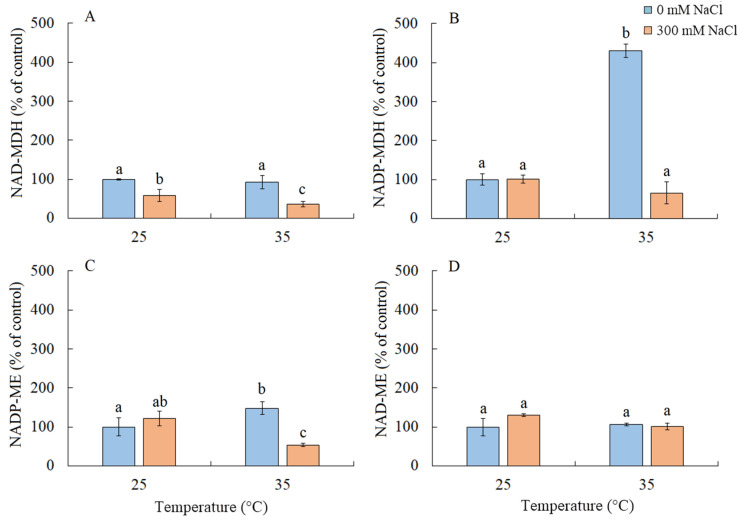
The effect of salinity (300 mM NaCl) and elevated temperature (35 °C) and their interaction with the malate dehydrogenase complex in *Sedobassia sedoides* shoots. The different letters show statistically different means at *p* ≤ 0.05 (Tukey test). (**A**) NAD-MDH; (**B**) NADP-MDH; (**C**) NADP-ME; (**D**) NAD-ME.

**Figure 8 plants-13-00800-f008:**
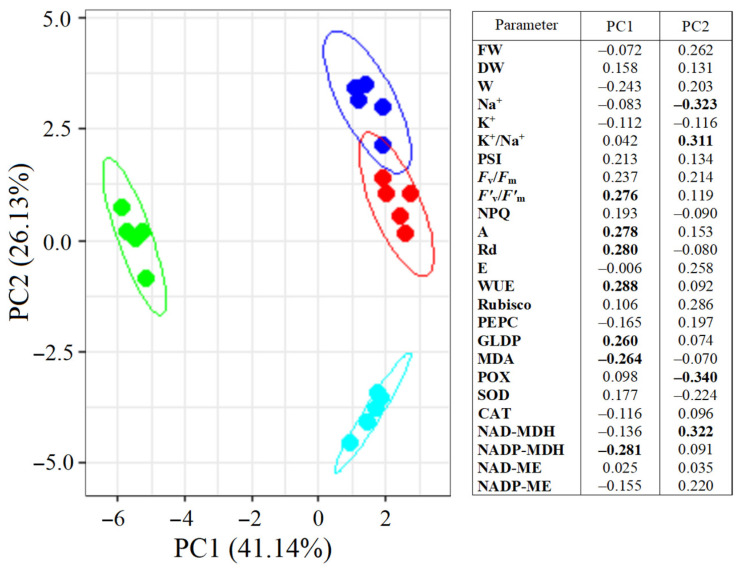
Principal component analysis (PCA) of the physiological data of *Sedobassia sedoides* plants under salinity (300 mM NaCl) and elevated temperature (35 °C). Dark blue: control (25 °C + 0 mM NaCl); red: NaCl (25 °C + 300 mM NaCl); green: eT (35 °C + 0 mM NaCl); blue: et + NaCl (35 °C + 300 mM NaCl). The main significant factors are bold.

**Figure 9 plants-13-00800-f009:**
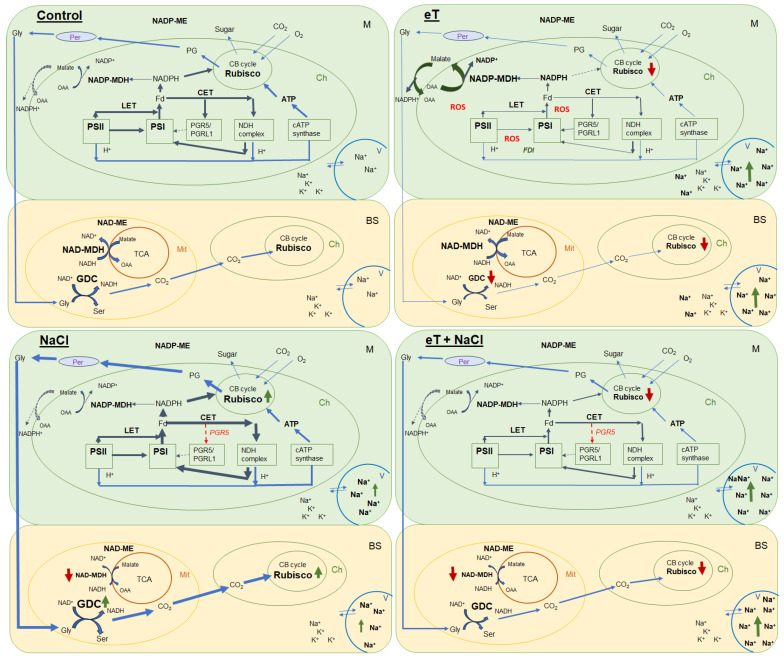
Schematic presentation of the changes in the photosynthetic reactions of the C_2_ halophyte *Sedobassia sedoides* in response to elevated temperature (35 °C) and salinity (300 mM NaCl), both individually and in combination. Control (25 °C + 0 mM NaCl); NaCl (25 °C + 300 mM NaCl); eT (35 °C + 0 mM NaCl); eT + NaCl (35 °C + 300 mM NaCl). M, mesophyll cell; BS, bundle sheath cell; Ch, chloroplast; Mt, mitochondria; Per, peroxisome; V, vacuole; CB cycle, Calvin-Benson cycle; TCA, tricarboxylic acid flux; LET, linear electron transport; CET, cyclic electron transport; PSI, photosystem I; PSII, photosystem II; GDC, glycine decarboxylase; Gly, glycine; Ser, serine; PG, phosphoglycolates; OAA, oxaloacetate; *FDI*, gene encoding ferredoxin I; *PGR5*, gene encoding the PGR5 protein, a key component of the main CET pathway of PSI. Red arrow indicates decreased activity/expression, and green arrow indicates increased content/activity.

**Table 1 plants-13-00800-t001:** List of used primers.

Primer	Gene	Function	5′–3′ Sequence
*rbcL*	AY270063.1	Large subunit (L) Rubisco	AGGCAAGCTTGAAGGGGAAA CTCCTGAAGCAACAGGCAGA
*Ppc2*	LOC110737782	PEPC isoform 2	GGAGGTGGACCTACCCATCT CTCAAGAGTGGCAGCAGTGA
*GLDP*	LOC110705444	P subunit GDC	TGTGGTTGATTTCAGCGCGT CGGCAACTAAGCCAGAAGGT
*PPDK*	MK674493.1	Pyruvate phosphate dikinase	GGTAAGGAATGAAACTAGCCCAGAGG GATCTCAGAGCACCCTGAAACACAAC
*psaA*	OK539756.1	Apoprotein A1 of photosystem I	CCGCGCCCGCTAAATAAAAA AATGGGTGGCTCCGTGATTT
*psaB*	LOC32958940	Apoprotein A2 of photosystem I	GAACCGCGTGCATCTAAAGC GCCTGGCTGGTTAAATGCTG
*psbA*	AY251266.1	Protein D1 of photosystem II	CAGGCTGAGCACAACATCCT AATAGGGAGCCGCCGAATAC
*FDI*	LOC110699227	Ferredoxin I protein	GAGTTTGAGTGCCCGGATGA CTGGTCGAGAGTACCAGACG
*PGR5*	LOC110692940	PGR5 protein, a key part of the main CET pathway of PSI	TCACAACCACAAGAGGAGCAA TCGCGTCCGGTGAGAATTAC
*NdhH*	LOC32959000	49 kDa subunit of the NADH dehydrogenase in the second CET pathway of PSI	GGCCATTTCACCGATTCGTA GGCCCTATGCTACGAGCTTC
*CAB*	LOC110735177	Chlorophyll a/b-binding protein LHCB/CAB PS II	TTCCAGGAGGTCAAGCAACC AGCTCCACCAGGGTACTTCT
*UBQ10*	LOC110721034	Ubiquitin 10 (reference gene)	CTTGTCCTTCGTCTCCGTGG CGCCATATACTTCACGCCGA
*b-Tubulin*	XM_021890176	b-tubulin (reference gene)	ACCGGAGAAGGTATGGACGA GTACTCTTCCTCATCGGCGG

## Data Availability

Data are contained within the article.

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
