# Peer review of "Salinity Mitigates the Negative Effect of Elevated Temperatures on Photosynthesis in the C3-C4 Intermediate Species Sedobassia sedoides"

_plants, 2024, doi:10.3390/plants13060800_

Round 1

Reviewer 1 Report

Comments and Suggestions for Authors

Comments paper: “Salinity mitigates the negative effect of elevated temperatures on photosynthesis in the C3-C4 intermediate species Sedobassia sedoides.”

Shuyskaya et al., 2024, Plants 2894263

The paper describes a study on the response of the C3-C4 halophyte Sedobassia sedoides on elevated temperature and salinity as separate stress factors and as combined. The study includes determination of water, Na+ and K+ content in terms of osmotic regulation, CO2/H2O gas exchange and chlorophyll fluorescence measurements to analyse the PSII function and the activity of cyclic electron transport around PSI, the amount and activity of some photosynthetic enzymes, RT-PCR analysis of some genes involved in the regulation of photosynthetic processes, and analysis of antioxidant and lipid peroxidation enzymes and of the malate metabolism.

The paper is well written, materials and methods are described in detail, the description of the results and the presentation of the figures has been taken care off. The strength of the work is the study of combined effect of two stress factors, here temperature and salinity. The authors showed that a simultaneous occurrence of stress factors can have less negative effects on the physiology of plants compared to the effect of the stress factors separately.

However, the manuscript can still be improved, especially some parts of the discussion should be revised:

p.9, first paragraph: The possible relation between PSI CET, PGR5 protein (and its gene) and C2CCM should be explained better. This is not clear.

p.10, fig. 9: Try to highlight more the essential parts of the figure, it is pretty hard to find, also because the figures are quite small. Simplify and/or rearrange the figures.

p.10, last part of the first paragraph, cfr. line 309: this is confusing, because in lines 254-255, you mention an increase in PSI CET activity, referring to the same figure 3. Make this more clearly: in the paragraph with lines 245-255, the effect of salinity is discussed, here it is the effect of temperature.  So, compare the two;

and next paragraph: the same here: from line 311 on: cfr lines 251-252: "stable maximum quantum yield of PSII", here “a significant decrease is PSSII efficiency”; the effect of salinity is discussed in lines 251-2, here (lines 311) it is the effect of temperature.  So, compare the two.

p.11: The relation with the effect of temperature is not clear to me.

In line 331-3 you write "high temperature increased water content and Na+ accumulation under non-saline conditions"; then you explain SOS and in lines 340-2: "perhaps the lack of sodium in the root environment..;", but in line 333, "non-saline conditions (i.e. 0.13 mM Na+/L) is present: this is confusing. Reformulate this part and make it more clear.

Author Response

We are very grateful to the reviewer for careful and detailed consideration of our article and for useful advices, comments and recommendations. We have thoroughly revised our manuscript in accordance with the reviewer’ recommendations. All changes are highlighted in green.

Comments paper: “Salinity mitigates the negative effect of elevated temperatures on photosynthesis in the C3-C4 intermediate species Sedobassia sedoides.” Shuyskaya et al., 2024, Plants 2894263

The paper describes a study on the response of the C3-C4 halophyte Sedobassia sedoides on elevated temperature and salinity as separate stress factors and as combined. The study includes determination of water, Na+ and K+ content in terms of osmotic regulation, CO2/H2O gas exchange and chlorophyll fluorescence measurements to analyse the PSII function and the activity of cyclic electron transport around PSI, the amount and activity of some photosynthetic enzymes, RT-PCR analysis of some genes involved in the regulation of photosynthetic processes, and analysis of antioxidant and lipid peroxidation enzymes and of the malate metabolism.

The paper is well written, materials and methods are described in detail, the description of the results and the presentation of the figures has been taken care off. The strength of the work is the study of combined effect of two stress factors, here temperature and salinity. The authors showed that a simultaneous occurrence of stress factors can have less negative effects on the physiology of plants compared to the effect of the stress factors separately.

However, the manuscript can still be improved, especially some parts of the discussion should be revised:

p.9, first paragraph: The possible relation between PSI CET, PGR5 protein (and its gene) and C2CCM should be explained better. This is not clear.

Response – Thank you for the comments. We tried to explain the possible relation between PSI CET, PGR5 protein (and its gene) and C2CCM (lines 258-266).

p.10, fig. 9: Try to highlight more the essential parts of the figure, it is pretty hard to find, also because the figures are quite small. Simplify and/or rearrange the figures.

 Response – Thank you for the recommendation.  We tried to simplify the figures.

p.10, last part of the first paragraph, cfr. line 309: this is confusing, because in lines 254-255, you mention an increase in PSI CET activity, referring to the same figure 3. Make this more clearly: in the paragraph with lines 245-255, the effect of salinity is discussed, here it is the effect of temperature.  So, compare the two;

Response – Thank you for the recommendation.  We have added this comparison (lines 308-210).

and next paragraph: the same here: from line 311 on: cfr lines 251-252: "stable maximum quantum yield of PSII", here “a significant decrease is PSSII efficiency”; the effect of salinity is discussed in lines 251-2, here (lines 311) it is the effect of temperature.  So, compare the two.

Response – Thank you for the recommendation.  We have added this comparison (lines 324-326).

p.11: The relation with the effect of temperature is not clear to me.

In line 331-3 you write "high temperature increased water content and Na+ accumulation under non-saline conditions"; then you explain SOS and in lines 340-2: "perhaps the lack of sodium in the root environment..;", but in line 333, "non-saline conditions (i.e. 0.13 mM Na+/L) is present: this is confusing. Reformulate this part and make it more clear.

Response – Thank you for the recommendation.  We reformulated this part (lines 346-348, 354).

Reviewer 2 Report

Comments and Suggestions for Authors

The manuscript by Shuyskaya et al studied the response of Sedobassia sedoides to elevated temperature, salinity, and their combined effect. The description of the experimental design, plant treatments, sample preparation, measurement methods, and data analysis is detailed. The interpretation of the data is appropriate. I would suggest publication of this manuscript.

Cheak line 120 with Fig 1A, and line 215-217 with Fig 7.

Author Response

We are very grateful to the reviewer for consideration of our article and comments. We have revised our manuscript in accordance with the reviewer’ recommendations. All changes are highlighted in green.

The manuscript by Shuyskaya et al studied the response of Sedobassia sedoides to elevated temperature, salinity, and their combined effect. The description of the experimental design, plant treatments, sample preparation, measurement methods, and data analysis is detailed. The interpretation of the data is appropriate. I would suggest publication of this manuscript.

Cheak line 120 with Fig 1A, and line 215-217 with Fig 7.

Response – We have corrected the links in Fig. 1 in text (line 121), and changed the order of the sub-figures in Fig. 7

Reviewer 3 Report

Comments and Suggestions for Authors

The author used a special C2 type species as material to study the effects of salt and high temperature on the growth and photosynthesis. The author observed that 300mM Nacl alleviated the damage caused by high temperature. This study is interesting and meaningful.

However, there are some issues that need to be resolved before publication.

Q1: Why didn't the author compare this C2 species with closely related C3 or C4 species? If so, the research will be more meaningful.

Q2: The measurement condition of gas exchange, chlorophyll fluorescence parameters should be described in detail, such as temperature.

Author Response

We are very grateful to the reviewer for consideration of our article and for useful advices and recommendations. We have thoroughly revised our manuscript in accordance with the reviewer’ recommendations. All changes are highlighted in green

The author used a special C2 type species as material to study the effects of salt and high temperature on the growth and photosynthesis. The author observed that 300mM Nacl alleviated the damage caused by high temperature. This study is interesting and meaningful.

However, there are some issues that need to be resolved before publication.

Q1: Why didn't the author compare this C2 species with closely related C3 or C4 species? If so, the research will be more meaningful.

 Response – Thank you for the recommendation.  We have added such comparison (lines 404-410).

Q2: The measurement condition of gas exchange, chlorophyll fluorescence parameters should be described in detail, such as temperature.

Response – Thank you for the recommendation.  We have expanded the description of these methods (lines 443-446, 477-478).

Round 2

Reviewer 1 Report

Comments and Suggestions for Authors

Dear authors,

Thank you for reviewed version of your manuscript in which you did take account of the comments I made. Therefore, I suggest that the paper can be accepted for publication.